# Dynamic Profiles of Fermentation Quality and Microbial Community of Kudzu (*Pueraria lobata*) Ensiled with Sucrose

Zhenping Hou † , Xia Zheng † , Xuelei Zhang, Li Yan, Qing Chen and Duanqin Wu *

Institute of Bast Fiber Crops, Chinese Academy of Agricultural Sciences, Changsha 410205, China
* Correspondence: wuduanqin@caas.cn
† These authors contributed equally to this work.

**Abstract:** The study aimed to investigate the effects of different levels of added sucrose on the fermentation quality and microbial community of kudzu (*Pueraria lobata*) silage. The three sucrose supplementation levels utilized were 0, 0.5, and 1.0%, and kudzu was silaged for 15, 30, and 60 days. Sucrose supplementation significantly decreased the pH levels, acid detergent fiber, ammonia nitrogen content, and relative abundance of *Pantoea* in the silages ($p < 0.05$). The addition of 1% sucrose to kudzu silage at 60 days had the lowest acid detergent fiber content, the highest crude protein, and the highest relative feed value. Additionally, the sucrose-supplemented silage had a lower pH than that of the control group at each time point. The dominant genera in all groups were *Klebsiella*, *Enterobacteriaceae*, *Lactobacillus*, and *Weissella*, and the relative abundance of *Enterobacteriaceae* was lower in the 1% sucrose-supplemented group than in the control group. These results showed that sucrose addition could improve the quality of kudzu silage and increase its beneficial microbial community.

**Keywords:** kudzu; silage; fermentation quality; microbial community





## 1. Introduction

Kudzu (*Pueraria lobata*) is a perennial, semi-woody, and leguminous vine with hairy rust-brown stems of the *Pueraria* genus. Although native to East Asia, it has spread worldwide and is predominantly found in temperate climates. Kudzu usage originated in China, with the oldest Chinese written reference to the use of kudzu described in the Classic of Poetry (Shih Ching). Kudzu is a rich source of polyphenolic compounds, including isoflavones, isoflavonoid glycosides, and coumarins, and presents abundant secondary metabolites such as puerarin, daidzin, and aglycone daidzein [1,2]. Kudzu, and especially its roots, is used to relieve fever, diarrhea, and vomiting [3], and it was first documented as a medicinal herb in the Divine Husbandman's Classic of Chinese Materia Medica (Shen Nong Ben Cao Jing). In addition to its functional active ingredients, kudzu stems and leaves are rich in nutrients and have a high protein content with good palatability; thus, it can be used as high-quality forage grass [4–6]. Kudzu has a longer green period in a year and a higher biological yield than most legume forage plants. Ensiling, a traditional fresh forage preservation method that utilizes microorganisms to degrade water-soluble carbohydrates (WSC) to produce organic acids, reduce pH, and inhibit the proliferation of decaying bacteria [7], is the best way to preserve roughage in high-temperate and rainy areas. As with other legumes, such as soybeans, alfalfa, and clover, it is difficult to produce lactic acid for silage because of the low WSC content in kudzu.

In recent years, there have been many studies on the effects of WSC addition on the silage of alfalfa, clover, and soybeans. Molasses, starch, fructose, and sucrose are the main sources of WSC that can improve the quality of legume forage silage [8,9]. Sucrose has been widely used to improve the quality of silage because it can increase the supply of substrates that lactic acid bacteria need to grow [10,11].

Different microbial types play key roles during the fermentation process during forage ensilage [12]. Therefore, monitoring the changes in the chemical and microbial

composition during ensiling can help improve our understanding of this process [12]. To identify the changes in microbial types during silage, 16S rDNA sequencing has been used to analyze the bacterial profiles of the present organisms [13]. This experiment was conducted to explore the effects of sucrose supplementation on the fermentation quality and microbial communities of kudzu silage, providing a scientific basis and guidance for forage application of kudzu.

## 2. Materials and Methods

### 2.1. Silage Preparation

Kudzu plants were cultivated in Yuanjiang (experimental station of the Institute of Bast Fiber Crops, Chinese Academy of Agricultural Sciences, Yuanjiang County, Hunan, China, latitude 28.82° N, longitude 112.3° E). Commercial sucrose (Jinan Feiyang Chemical Co., Ltd., Jinan, China) was used as an additive mixed with kudzu before ensiling. In July 2020, fresh kudzu stems and leaves were harvested, mechanically chopped into approximately 2 cm long pieces using a straw chopper, and wilted until the dry matter (DM) dropped to 65%. The kudzu stem and leaf stocks were then mixed and randomly assigned to three groups: the PL (control, no added sucrose), PS1 (0.5% added sucrose), and PS2 (1.0% added sucrose) groups.

The mixtures (approximately 10 kg) were placed in 36 vacuum polyethylene plastic bags (25 × 40 cm, Shandong Junde Biological Technology Co., Ltd., Zhucheng, China) and then vacuum-sealed after being thoroughly mixed. The bags were maintained at room temperature (23–27 °C) for 15, 30, and 60 d. Three bags were randomly selected for each treatment at each time point. Silage samples were then transferred from the bags to plastic beakers for homogeneous mixing and used for analysis.

### 2.2. Fermentation Quality Analysis

To determine fermentation quality, the pre-ensiled kudzu and silage samples were dried to a constant weight at 65 °C for 72 h, then ground into 40 meshes. Dry matter (DM) and crude protein (CP) content were analyzed according to the methods of the Association of Official Analytical Chemists [14]. The quantitative analysis of fiber composition (crude fiber (CF), neutral detergent fiber (NDF), and acid detergent fiber (ADF)) was performed according to the method described by Vansoest et al. [15] using an Ankom fiber analyzer (F800, Haineng, Jinan Haineng Instrument Technology Co., LTD, Jinan, China). Gross energy (GE) was measured according to the bomb calorimeter method described by the International Organization for Standardization (ISO 9831-1998) using an isothermal auto-calorimeter (5E-C5508, Changsha Kaiyuan Instrument Co., Ltd., Changsha, China).

Fresh silage samples (20 g each) were blended overnight with 180 mL of distilled water at 4 °C and then filtered through four layers of cheesecloth and qualitative filter paper. The pH of the filtrates was immediately measured using a digital S210-Basic pH meter (METTLER TOLEDO Solutions, Shanghai Precision & Scientific Instrument Co., LTD, Shanghai, China). The ammonia nitrogen ($NH_3$-N) content in the water extract was determined according to the method reported by Weatherburn [16], using an ELx808 spectrophotometer (BioTek Instruments, Inc., Winooski, VT, USA).

### 2.3. Relative Feed Value Evaluation

The relative feed value (RFV) indices of kudzu silage were expressed as dry matter intake (DMI, % body weight) and digestible dry matter (DDM, % DM), which were calculated using ADF (%) and NDF (%), respectively [17] The RFVs of silages were predicted as follows: DMI (%) = 120/NDF, DDM (%) = 88.9–0.779 × ADF, RFV = DMI × DDM × 0.775.

### 2.4. Bacterial Community Analysis

Total DNA was extracted using the E.Z.N.A. ®Stool DNA Kit (D4015, Omega, Inc., Norcross, GA, USA), according to the manufacturer's instructions. After DNA purification and concentration determination, the V3-V4 region of the bacterial 16S rDNA

gene was amplified with the primers 341F (5′-CCTACGGGNGGCWGCAG-3′) and 806R (5′-GGACTACHVGGGTWTCTAAT-3′) using a GeneAmp 9700 thermocycler (ABI Inc., Edison, NJ, USA) [18]. The polymerase chain reaction (PCR) mixtures (25 μL) contained 25 ng of DNA template, 12.5 μL of PCR Premix, 2.5 μL of primer 341F, 2.5 μL of primer 806R, and PCR-grade water to adjust the volume. The PCR conditions were as follows: initial denaturation at 98 °C for 5 min; 35 cycles of denaturation (98 °C, 10 s), annealing (54 °C, 30 s), and extension (72 °C, 45 s); final extension at 72 °C for 10 min. Amplicon pyrosequencing was conducted using an Illumina MiSeq platform, according to the manufacturer's (Illumina, Inc., Lianchuan Biotechnology Co., Ltd., Hangzhou, China). When the average mass fraction of the 10 bp sliding window was lower than 20, the paired end of the sample was read according to the unique barcode assignment and truncated by cutting the bar code and primer sequence. Low-quality sequences that contained undetected nucleotides or were shorter than 200 bp were filtered out using the bVsearch software (v2.3.4) [19]. Quality filtering of the raw tags was performed according to FQTRIM (version 0.94) to obtain high-quality clean tags. Chimeric sequences were filtered using the Vsearch software (version 2.3.4), and sequences with similarities equal to or greater than 97% were assigned to the same operational taxonomic units (OTUs). The QIIME software (version 1.8.0) was used to calculate the sample diversity. Several alpha diversity indices (OUT, Chao1, observed species, Good's coverage, Shannon, and Simpson's) were used to analyze the samples species diversity complexity. Beta diversity analysis was performed using principal coordinate analysis (PCoA) and the QIIME software (version 1.8.0) cluster analysis, used to evaluate species complexity differences among samples. Spearman's correlation analysis (SPSS, version 21.0, Chicago, IL, USA) was used to calculate correlations between the main genera and silage quality. All sequences in this study were deposited in the sequence read archive of the NCBI database under accession number PRJNA855559.

*2.5. Statistical Analysis*

Statistical differences were evaluated using two-factor ANOVA. The fixed effects were sucrose treatments, ensilage duration, and sucrose treatment interaction with ensilage duration, which were determined using the SAS general linear model (version 8.0; SAS Institute, Inc., Cary, NC, USA). All data are represented as the mean. The Tukey–Kramer multiple comparison test was used to determine the differences between the means. $p < 0.05$ was considered significant, $p < 0.01$ was considered extremely significant, and $0.05 \leq p < 0.1$ was considered as a tendency of significant difference. All figures were generated using the GraphPad Prism software (version 5.0, La Jolla, CA, USA).

## 3. Results

*3.1. Chemical Composition of the Kudzu Silages*

The DM of the kudzu silage was significantly affected by the ensilage duration (D) and the interaction between the sucrose treatment and the ensilage duration (I) (Table 1). The DM content was significantly higher ($p < 0.05$) on day 60 of ensilaging compared to day 30 in the PL group, and compared to days 15 and 30 in the PS1 group. The CP content of the kudzu silage was significantly affected ($p < 0.001$) by the duration of the ensilage, as it was significantly higher ($p < 0.001$) on day 60 than on day 30. The ADF content of the kudzu silage was significantly affected ($p < 0.05$) by sucrose addition (T) and ensiling duration. The ADF content was comparable among the three groups on days 15 and 60, but there was a significantly higher ($p < 0.05$) ADF content in the silages of the PL and PS1 groups than in that of the PS2 group on day 30. The lowest ADF content was recorded in PS2 silage at day 60, with a value of 324.83 g/kg. The CF content of the kudzu silage was significantly affected by the duration of ensiling ($p < 0.01$), and the highest CF content was recorded on day 30. After 60 days of ensilaging, the lowest CF content was 247.27 g/kg DM in the PS1 group. Sucrose addition, ensilaging duration, and their interactions had no effect ($p > 0.05$) on NDF and GE content (Table 1).

**Table 1.** Chemical compositions of the kudzu silages (g/kg DM).

| Items | Treatment | Days Ensiled | | | SEM | p-Value | | |
|---|---|---|---|---|---|---|---|---|
| | | **15** | **30** | **60** | | **T** | **D** | **I** |
| DM | PL | 238.03 ab | 236.13 b | 246.10 a | 5.35 | ns | * | * |
| | PS1 | 243.00 b | 234.23 b | 275.73 a | | | | |
| | PS2 | 252.33 | 242.93 | 241.80 | | | | |
| CP | PL | 145.83 b | 142.75 b | 151.65 a | 3.86 | ns | *** | ns |
| | PS1 | 144.50 b | 138.13 b | 157.58 a | | | | |
| | PS2 | 146.17 ab | 138.22 b | 160.73 a | | | | |
| NDF | PL | 523.70 | 501.43 | 487.93 | 10.81 | ns | ns | ns |
| | PS1 | 489.77 | 500.50 | 476.23 | | | | |
| | PS2 | 475.30 | 473.97 | 477.30 | | | | |
| ADF | PL | 338.90 b | 361.33 $^{\alpha}$a | 335.33 b | 5.24 | * | ** | ns |
| | PS1 | 350.30 b | 369.90 $^{\alpha}$a | 328.87 c | | | | |
| | PS2 | 333.97 | 334.43 $^{\beta}$ | 324.83 | | | | |
| CF | PL | 284.47 b | 295.73 a | 250.10 c | 10.97 | ns | ** | ns |
| | PS1 | 294.47 b | 314.10 a | 247.27 c | | | | |
| | PS2 | 263.40 b | 292.20 a | 256.93 b | | | | |
| GE | PL | 17.07 | 16.45 | 17.31 | 0.18 | ns | ns | ns |
| | PS1 | 16.88 | 16.74 | 17.06 | | | | |
| | PS2 | 16.61 | 16.40 | 16.78 | | | | |

DM, dry matter; CP, crude protein; NDF, neutral detergent fiber; ADF, acid detergent fiber; CF, crude fiber; GE, total energy; PL, 0% sucrose; PS1, 0.5% sucrose; PS2, 1.0% sucrose; SEM, standard error of the mean; T, sucrose treatment; D, ensiling duration; I, interaction between ensiling duration and sucrose treatment. The data were calculated based on the estimation formulas. [a]; [b]; [c] Means within a row with different superscripts differ ($p < 0.05$). $^{\alpha}$; $^{\beta}$ Means within a column with different superscripts differ significantly ($p < 0.05$). * $p < 0.05$; ** $p < 0.01$; *** $p < 0.001$; ns, not significant.

### 3.2. Fermentation Dynamics Profile of Kudzu Silage

The results of fermentation characteristics of the kudzu silage with or without sucrose are shown in Table 2. Sucrose addition significantly affected the pH ($p < 0.001$), with the silage in the PL group maintaining the highest pH value. The pH values of the kudzu silage in the PS1 and PS2 groups on days 30 and 60 were similar but significantly lower ($p < 0.001$) than that of the PL group. The minimum pH value was 4.54 (observed in the PS2 silage on day 60) and was significantly lower ($p < 0.001$) than that in the PL silage.

**Table 2.** pH and $NH_3$-N content of different kudzu silages.

| Items | Treatment | Days Ensiled | | | SEM | p-Value | | |
|---|---|---|---|---|---|---|---|---|
| | | **15** | **30** | **60** | | **T** | **D** | **I** |
| pH | PL | 5.15 | 5.34 $^{\alpha}$ | 5.24 $^{\alpha}$ | 0.12 | *** | ns | ns |
| | PS1 | 4.90 | 5.09 $^{\alpha\beta}$ | 4.59 $^{\beta}$ | | | | |
| | PS2 | 4.68 | 4.78 $^{\beta}$ | 4.54 $^{\beta}$ | | | | |
| $NH_3$-N (g/kg TN) | PL | 11.58 b | 14.13 a | 11.85 b | 0.45 | ns | *** | ns |
| | PS1 | 12.33 b | 14.74 a | 12.69 b | | | | |
| | PS2 | 12.09 b | 15.42 a | 10.81 b | | | | |

$NH_3$-N, ammonia nitrogen; TN, total nitrogen; PL, 0% sucrose; PS1, 0.5% sucrose; PS2, 1.0% sucrose; SEM, standard error of the mean; T, sucrose treatment; D, ensiling duration; I, interaction between ensiling duration and sucrose treatment. [a]; [b] Means within a row with different superscripts differ ($p < 0.05$). $^{\alpha}$; $^{\beta}$ Means within a column with different superscripts differ significantly ($p < 0.05$). *** $p < 0.001$; ns, not significant.

The $NH_3$-N content was significantly affected by ensiling duration ($p < 0.001$). In the early and final phases of ensiling, the $NH_3$-N content was comparable among the three groups ($p > 0.05$); the $NH_3$-N content in the PL, PS1, and PS2 silages was highest on day 30, and it was significantly higher ($p < 0.001$) than those of the silages on days 15 and 60.

### 3.3. Assessment of Relative Feeding Value

The sucrose treatment, ensiling duration, and the interaction between the two did not significantly affect ($p > 0.05$) the DMI values in any of the silages (Table 3). Ensiling duration significantly affected ($p < 0.01$) the DDM values, which were comparable among the three groups on days 15 and 30, and significantly increased ($p < 0.01$) in the PS1 silage on day 60. The DDM reached its highest value (63.60% DM) in the PS1 silage on day 60, and it did not significantly change ($p > 0.05$) in the PL and PS2 groups with increasing ensiling duration. The RFV values of the silages were not significantly affected ($p > 0.05$) by sucrose treatment, ensiling duration, or their interaction.

**Table 3.** Relative feed values of kudzu silage.

| Items | Treatment | Days Ensiled | | | SEM | p-Value | | |
|---|---|---|---|---|---|---|---|---|
| | | **15** | **30** | **60** | | **T** | **D** | **I** |
| DMI% | PL | 2.30 | 2.39 | 2.47 | 0.05 | ns | ns | ns |
| | PS1 | 2.45 | 2.40 | 2.52 | | | | |
| | PS2 | 2.53 | 2.53 | 2.52 | | | | |
| DDM% | PL | 62.50 | 60.75 | 62.78 | 0.41 | ns | ** | ns |
| | PS1 | 61.61 [ab] | 60.08 [b] | 63.60 [a] | | | | |
| | PS2 | 62.89 | 62.85 | 63.28 | | | | |
| RFV | PL | 111.55 | 112.74 | 120.23 | 2.98 | ns | ns | ns |
| | PS1 | 117.03 | 111.73 | 123.78 | | | | |
| | PS2 | 123.26 | 123.46 | 124.23 | | | | |

DMI, dry matter intake; DDM, digestible dry matter; RFV, relative feed value; PL, 0% sucrose; PS1, 0.5% sucrose; PS2, 1.0% sucrose; SEM, standard error of the mean; T, sucrose treatment; D, ensiling duration; I, interaction between ensiling duration and sucrose treatment. The data were calculated based on the estimation formulas. [a]; [b] Means within a row with different superscripts differ. ** $p < 0.01$; ns, not significant.

### 3.4. Microbial Community Diversity in Kudzu Silage

The results of the alpha diversity analysis of the kudzu silage on days 15, 30, and 60 are presented in Table 4. The sucrose treatment, ensiling duration, and their interaction did not significantly affect the OTUs, observed species, Chao1, Shannon, Simpson, and coverage indices of the kudzu silages. However, ensiling duration had a tendency ($p = 0.09$) to affect the Simpson indices of the kudzu silage, and the silage in the PS2 group on day 60 had the greatest value (0.84). The interaction between sucrose treatment and ensiling duration had a tendency ($p = 0.06$) to influence the coverage of the kudzu silage.

PCoA, which was unweighted UniFrac-based, showed a distinct clustering of the macrobiotic compositions for each group (R = 0.3601, $p = 0.001$) (Figure 1A). Principal coordinates 1 and 2 accounted for 14.33% and 8.35% of the total variance, respectively (Figure 1A). Taxonomic analysis showed that the microbiota of the kudzu silage was represented by four phyla: *Proteobacteria*, *Firmicutes*, *Cyanobacteria*, and *Actinobacteria* (Figure 1B and Table 5). The relative abundances of *Proteobacteria* and *Firmicutes* were significantly affected ($p < 0.001$) by the interaction between sucrose addition and ensiling duration. The relative *Proteobacteria* reached its highest value, while *Firmicutes* reached its lowest value in the PS2 group after day 60 of ensiling (Table 5). The relative abundance of *Cyanobacteria* was significantly affected ($p < 0.001$) by ensiling duration, as it was significantly higher ($p < 0.001$) on day 60 of the kudzu silage than those on days 15 and 30; moreover, the silage in the PS2 group on day 60 had the greatest relative abundance of *Cyanobacteria* (4.3) among all the silages (Table 5). The relative abundance of *Actinobacteria* was significantly affected ($p < 0.05$) by ensiling duration and its interaction with sucrose addition. In the PL group, the relative abundance of *Actinobacteria* on day 15 of the kudzu silage was significantly higher ($p < 0.05$) than those on days 30 and 60 (Table 5). In the PS1 and PS2 groups, the relative abundance of *Actinobacteria* was highest on day 60, which was significantly higher ($p < 0.05$) than those on days 15 and 30. The silage in the PS2 group on day 60 had the highest relative abundance of *Actinobacteria* (3.55) among all the silages (Table 5).

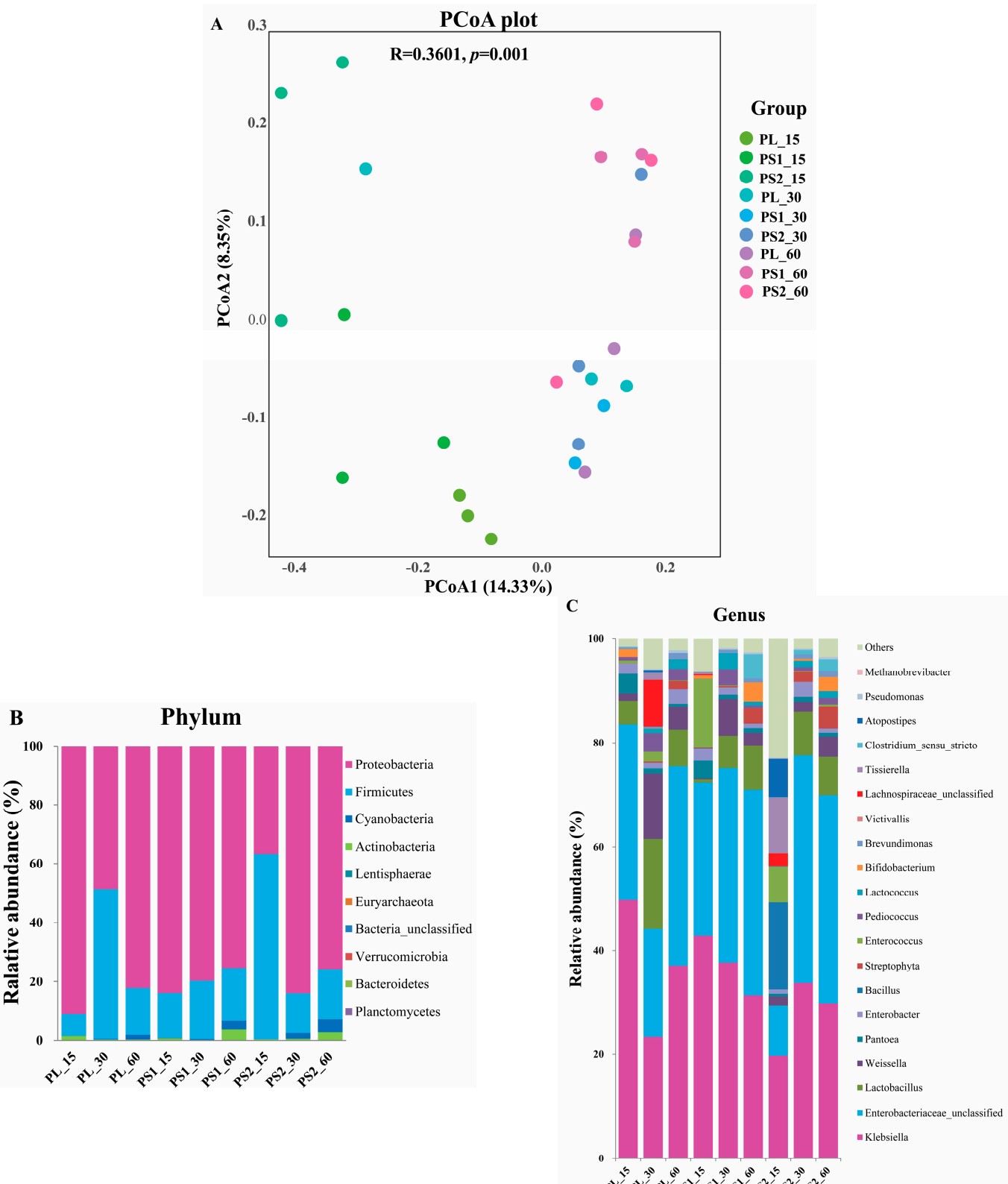

**Figure 1.** (**A**) Principal coordinate analysis (PCoA). (**B**) Relative abundance of the microbiota at the phylum level in the kudzu silage samples. (**C**) Relative abundance of the microbiota at the genus level.

**Table 4.** Alpha diversity of the bacterial communities of the kudzu silage.

| Items | Treatment | Days Ensiled | | | SEM | p-Value | | |
|---|---|---|---|---|---|---|---|---|
| | | 15 | 30 | 60 | | T | D | I |
| OTUs | PL | 692.67 | 498.33 | 633.67 | 71.87 | ns | ns | ns |
| | PS1 | 643.00 | 521.33 | 467.67 | | | | |
| | PS2 | 575.67 | 646.00 | 494.67 | | | | |
| Observed species | PL | 468.33 | 363.67 | 471.00 | 38.86 | ns | ns | ns |
| | PS1 | 462.67 | 432.67 | 471.00 | | | | |
| | PS2 | 390.00 | 545.33 | 427.67 | | | | |
| Chao1 | PL | 841.92 | 648.36 | 797.85 | 53.44 | ns | ns | ns |
| | PS1 | 852.67 | 810.56 | 729.29 | | | | |
| | PS2 | 651.68 | 884.47 | 744.11 | | | | |
| Shannon | PL | 3.47 | 3.79 | 3.86 | 0.28 | ns | ns | ns |
| | PS1 | 3.70 | 3.79 | 3.94 | | | | |
| | PS2 | 4.35 | 4.28 | 4.11 | | | | |
| Simpson | PL | 0.72 | 0.83 | 0.80 | 0.02 | ns | ns | ns |
| | PS1 | 0.76 | 0.80 | 0.82 | | | | |
| | PS2 | 0.83 | 0.83 | 0.84 | | | | |
| Coverage | PL | 0.97 | 0.98 | 0.98 | 0.00 | ns | ns | ns |
| | PS1 | 0.97 | 0.98 | 0.98 | | | | |
| | PS2 | 0.98 | 0.97 | 0.98 | | | | |

OTUs, operational taxonomic units; PL, 0% sucrose; PS1, 0.5% sucrose; PS2, 1.0% sucrose; SEM, standard error of the mean; T, sucrose treatment; D, ensiling duration; I, interaction between ensiling duration and sucrose treatment. The data were calculated based on the estimation formulas. ns, not significant.

**Table 5.** Relative abundance of the major phyla (%) present in silage samples.

| Items | Treatment | Days Ensiled | | | SEM | p-Value | | |
|---|---|---|---|---|---|---|---|---|
| | | 15 | 30 | 60 | | T | D | I |
| *Proteobacteria* | PL | 90.89 $^{\alpha a}$ | 48.70 $^{\beta b}$ | 82.21 $^{ab}$ | 5.24 | ns | ns | *** |
| | PS1 | 83.88 $^{\alpha}$ | 79.75 $^{\alpha\beta}$ | 75.36 | | | | |
| | PS2 | 36.72 $^{\beta b}$ | 75.71 $^{\alpha ab}$ | 84.00 $^{a}$ | | | | |
| *Firmicutes* | PL | 7.54 $^{c}$ | 50.75 $^{\alpha a}$ | 15.89 $^{b}$ | 4.79 | ns | ns | *** |
| | PS1 | 15.24 | 19.78 $^{\beta}$ | 17.74 | | | | |
| | PS2 | 62.86 $^{a}$ | 16.93 $^{\beta b}$ | 13.45 $^{b}$ | | | | |
| *Cyanobacteria* | PL | 0.04 $^{b}$ | 0.24 $^{b}$ | 1.59 $^{a}$ | 0.70 | ns | *** | ns |
| | PS1 | 0.13 $^{b}$ | 0.27 $^{b}$ | 2.92 $^{a}$ | | | | |
| | PS2 | 0.02 $^{c}$ | 1.94 $^{b}$ | 4.30 $^{a}$ | | | | |
| *Actinobacteria* | PL | 1.51 $^{a}$ | 0.22 $^{b}$ | 0.17 $^{\beta b}$ | 0.58 | ns | * | ** |
| | PS1 | 0.70 $^{b}$ | 0.11 $^{b}$ | 2.90 $^{\alpha\beta a}$ | | | | |
| | PS2 | 0.33 $^{b}$ | 0.53 $^{b}$ | 3.55 $^{\alpha a}$ | | | | |

PL, 0% sucrose; PS1, 0.5% sucrose; PS2, 1.0% sucrose; SEM, standard error of the mean; T, sucrose treatment; D, ensiling duration; I, interaction between ensiling duration and sucrose treatment. The data were calculated based on the estimation formulas. [a]; [b]; [c] Means within a row with different superscripts differ ($p < 0.05$). [α]; [β] Means within a column with different superscripts differ significantly ($p < 0.05$). * $p < 0.05$; ** $p < 0.01$; *** $p < 0.001$; ns, not significant.

Eleven major genera were found in the kudzu silages after ensiling for 60 days. *Klebsiella*, *Enterobacteriaceae*, *Lactobacillus* and *Weissella* were dominant in the kudzu silages, followed by *Pantoea*, *Enterobacter*, *Streptophyta*, *Enterococcus*, *Pediococcus*, *Lactococcus* and *Brevundimonas* (Figure 1C and Table 6). The relative abundance of *Klebsiella* was significantly affected ($p < 0.01$) by the interaction between sucrose addition and ensiling duration. The relative abundances of *Enterobacteriaceae* and *Lactococcus* were significantly affected ($p < 0.01$) by ensiling duration and its interaction with sucrose addition. The relative abundances of *Lactobacillus*, *Streptophyta*, *Enterococcus*, *Pediococcus*, and *Brevundimonas* were significantly affected ($p < 0.05$) by ensiling duration. The relative abundance of *Pantoea* was significantly affected ($p < 0.05$) by sucrose addition, ensiling duration, and their interaction; it decreased significantly ($p < 0.05$) with increasing ensiling duration,

and it was significantly lower ($p < 0.05$) in the PS2 group than those in the PL and PS1 groups on day 15 (Table 6).

**Table 6.** Relative abundance of genera (%) present in kudzu silage.

| Items | Treatment | Days Ensiled | | | SEM | *p*-Value | | |
|---|---|---|---|---|---|---|---|---|
| | | 15 | 30 | 60 | | T | D | I |
| *Klebsiella* | PL | 49.73 α | 23.44 | 37.17 | 4.15 | ns | ns | ** |
| | PS1 | 42.79 α | 37.74 | 31.44 | | | | |
| | PS2 | 19.70 β | 33.84 | 29.91 | | | | |
| *Enterobacteriaceae* | PL | 38.34 a | 33.84 a | 20.80 αb | 2.59 | ns | *** | *** |
| | PS1 | 39.49 a | 37.39 a | 29.62 αb | | | | |
| | PS2 | 43.73 a | 39.96 a | 9.78 βb | | | | |
| *Lactobacillus* | PL | 4.58 b | 7.31 a | 7.07 ab | 3.20 | ns | * | ns |
| | PS1 | 0.59 b | 6.25 a | 8.46 a | | | | |
| | PS2 | 0.04 b | 8.43 a | 7.46 a | | | | |
| *Weissella* | PL | 1.41 | 7.57 | 4.46 | 2.55 | ns | ns | ns |
| | PS1 | 0.21 | 7.15 | 2.59 | | | | |
| | PS2 | 1.71 | 1.94 | 3.92 | | | | |
| *Pantoea* | PL | 3.81 αa | 0.98 b | 0.56 b | 0.64 | * | * | ** |
| | PS1 | 3.35 αa | 0.84 b | 0.88 b | | | | |
| | PS2 | 0.52 β | 1.00 | 0.75 | | | | |
| *Enterobacter* | PL | 1.99 | 1.05 | 2.77 | 0.53 | ns | ns | ns |
| | PS1 | 2.28 | 1.35 | 0.84 | | | | |
| | PS2 | 0.83 | 2.86 | 0.80 | | | | |
| *Streptophyta* | PL | 0.04 b | 0.24 ab | 1.59 a | 0.70 | ns | ** | ns |
| | PS1 | 0.13 b | 0.27 b | 2.92 a | | | | |
| | PS2 | 0.02 b | 1.94 ab | 4.30 a | | | | |
| *Enterococcus* | PL | 0.48 | 1.86 | 0.10 | 2.99 | ns | * | ns |
| | PS1 | 13.30 a | 0.11 b | 0.17 b | | | | |
| | PS2 | 6.84 a | 0.11 b | 0.36 b | | | | |
| *Pediococcus* | PL | 0.69 b | 3.64 a | 2.23 a | 0.89 | ns | * | ns |
| | PS1 | 0.03 b | 3.17 a | 0.46 b | | | | |
| | PS2 | 0.03 | 0.81 | 1.27 | | | | |
| *Lactococcus* | PL | 0.02 b | 0.89 βab | 1.90 a | 0.26 | ns | *** | ** |
| | PS1 | 0.02 b | 3.05 αa | 0.76 ab | | | | |
| | PS2 | 0.04 b | 1.22 αβab | 1.32 a | | | | |
| *Brevundimonas* | PL | 0.21 | 0.29 | 1.12 | 0.28 | ns | * | ns |
| | PS1 | 0.02 | 0.63 | 0.79 | | | | |
| | PS2 | 0.05 b | 0.81 ab | 1.13 a | | | | |

PL, 0% sucrose; PS1, 0.5% sucrose; PS2, 1.0% sucrose; SEM, standard error of the mean; T, sucrose treatment; D, ensiling duration; I, interaction between ensiling duration and sucrose treatment. The data were calculated based on the estimation formulas. a; b Means within a row with different superscripts differ ($p < 0.05$). α; β Means within a column with different superscripts differ significantly ($p < 0.05$). * $p < 0.05$; ** $p < 0.01$; *** $p < 0.001$; ns, not significant.

### 3.5. Relationship between Genera and Silage Quality

Spearman correlation analysis showed that *Streptophyta* was negatively correlated with ADF ($r = -0.38$, $p < 0.05$) and CF ($r = -0.42$, $p < 0.05$), but positively correlated with DMD ($r = 0.38$, $p < 0.05$). *Pantoea* was positively correlated with CF ($r = 0.39$, $p < 0.05$). *Enterococcus* was negatively correlated with $NH_3$-N ($r = -0.43$, $p < 0.05$) (Figure 2).

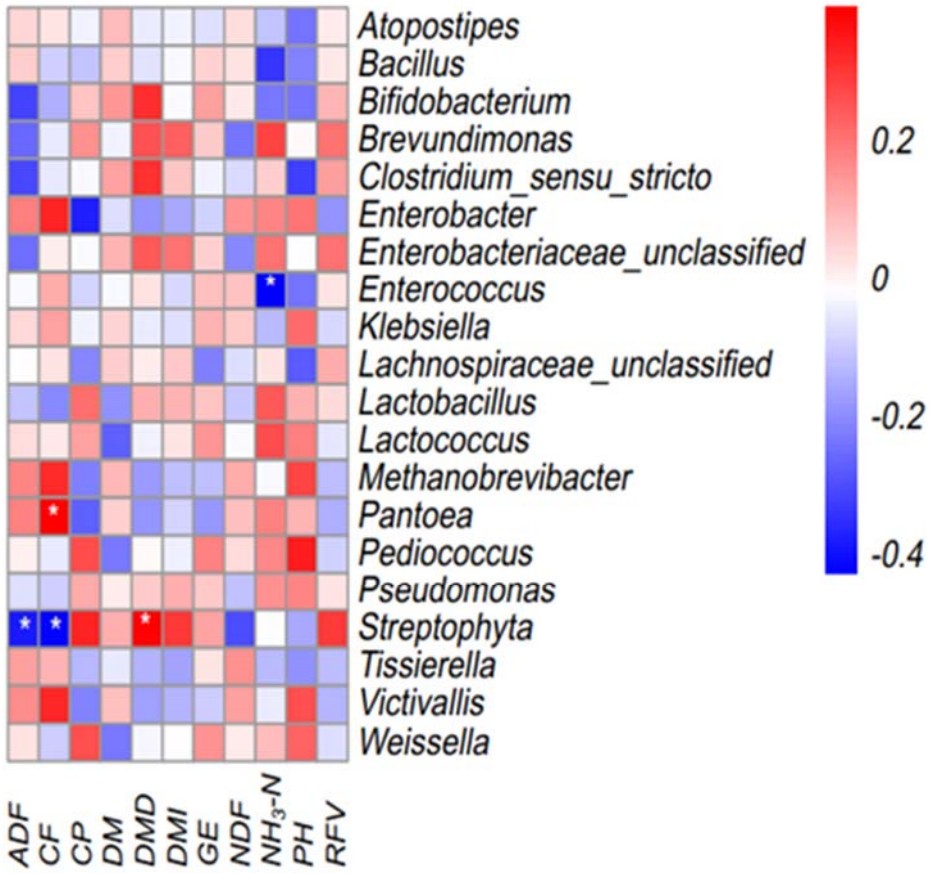

**Figure 2.** Spearman correlation heatmap between the main genera present in the kudzu silage and its quality. R was presented in different colors, with the right side of the legend showing the color range of the different R values. The value of $p \leq 0.05$ is marked with "*".

## 4. Discussion

It has been previously reported that the lower the pH value, the lower the $NH_3$-N content and the better the quality of the silage [20]; thus, pH is a key factor affecting silage quality [7]. In this study, the pH values in the silages first increased and then decreased with increasing ensiling duration. In the PS2 group, the pH value declined to a significantly lower level than that in the PL group at the early ensiling stage and remained at a stable low pH level during ensiling. The significantly lower pH value in the PS2 silage in this study indicates that pH inhibited the growth of harmful bacteria, and the addition of sucrose could improve the quality of silage fermentation products. This is consistent with the results of Li et al. [21], who found that the pH value of the alfalfa silage decreased after 120 d of fermentation when supplementing it with 0.67% sucrose. RFV is a comprehensive reflection of NDF and ADF in forage and is an important index for evaluating forage [17]. In this study, the addition of sucrose did not significantly affect the NDF content of kudzu silages in any group, but it significantly affected the ADF content, which was consistent with the result that there was a tendency ($p = 0.08$) for sucrose addition to affect RFV values of the silages. The silage in the PS2 group always maintained high RFV values among all the silages, even showing the highest RFV value (124.23) on day 60. This result showed that ADF had a greater influence on the forage RFV than NDF, and the addition of 1% sucrose had the potential to improve the feeding quality of the kudzu silage.

A higher $NH_3$-N content indicates greater protein breakdown and poor silage quality. In this investigation, the $NH_3$-N content showed the same tendency as the pH value, which increased and then decreased as the fermentation time increased. The lowest content of $NH_3$-N in the PS2 silage was 10.81 on day 60, which was significantly lower than those in the PL and PS1 silages on day 30. This result indicated that the addition of sucrose

contributed to the preservation of proteins, which is consistent with the results of Li et al., who found that sucrose addition could decrease the ammonia-N content in silage [22]. This is well illustrated by the CP values obtained in this study. After 60 days of silage fermentation, the CP content in the PS2 silage was highest, and it was significantly higher than those in the PL and PS1 groups. These results are consistent with the research results of Kang et al. [23] who found that the addition of 1% sucrose to alfalfa after 60 d of silage fermentation significantly decreased the $NH_3$-N content and significantly increased the CP content.

Li et al. [21] found that the addition of 0.67% sucrose could increase the DM content of alfalfa silage after 120 days of fermentation. In this study, we found that after 60 days of silage fermentation, the addition of sucrose had no significant effect on the DM content of the kudzu silage, but the DM content of the PS1 silage was highest on day 60 (275.73). These results indicated that the DM content in the silage could be increased after a certain period of fermentation with the addition of the appropriate amount of sucrose. Queiroz et al. [24] reported that the production of $NH_3$-N can be promoted by *Enterobacteriaceae*. In this study, the $NH_3$-N content of the PS2 silage was lowest on day 60, which was consistent with the fact that the *Enterobacteriaceae* community in the PS2 group was significantly lower than those in the PL and PS1 groups after 60 days of silage fermentation.

The present study results showed that the average coverage for all microbial DNA samples was nearly 98%, which suggests that the depth of sequencing was adequate for a reliable analysis of the bacterial community. In this study, the number of OTUs, observed species, and the Chao1, Shannon, and Simpson indexes were used to estimate the alpha diversity of the bacteria. Sucrose addition, ensiling time, and their interaction had no significant effect on alpha diversity in the present study, indicating that sucrose supplementation did not influence the richness and evenness of the bacterial communities in the kudzu silage fermentation product. The result from this study is contrary to that of Kang et al. [23], who found that ensiling time significantly affected the alpha diversity and that sucrose supplementation could stabilize the richness and evenness of the bacterial communities in the initial and terminal stages of alfalfa silage. This may be because they used alfalfa with sucrose for silage fermentation, whereas we used kudzu instead.

Our PCoA results showed that sucrose supplementation had an effect (R = 0.36, *p* = 0.001) on microbial composition. This result is similar to those of Kang et al. [23] and McGarvey et al. [25], who found that there was a difference in bacterial numbers after alfalfa silage fermentation, but bacterial diversity changed very little. This may be because changes at the level of some groups are offset by opposite changes at the level of others.

After 60 days of silage fermentation, the dominant phyla in the kudzu silages with or without sucrose were *Proteobacteria* and *Firmicutes*. Previous research has shown that the main phyla in the alfalfa silage with or without urea supplementation after 60 days of silage fermentation were also *Proteobacteria* and *Firmicutes* [26], which is further consistent with the results of other studies [23,27]. This may be because the low pH and anaerobic conditions of silage are more conducive to the growth of these two phyla. *Proteobacteria* and *Firmicutes* both showed significant and irregular changes with the increase in sucrose addition and the extension of ensiling duration. *Proteobacteria* has a low abundance in the gut of healthy humans, which has been suggested as a potential diagnostic criterion for ecological disorders and diseases [28]. After adding 1% sucrose, the structure of the bacterial community was significantly transferred during silage fermentation, and significantly higher *Proteobacteria* and lower *Firmicutes* populations were observed in the PS2 group than that in the PL group after 60 d of silage fermentation, contrary to the results of other studies [23,25,29,30]. This may be due to the different fermentation methods used.

At the genus level, *Enterococcus* and *Lactobacillus* normally play a dominant role in the fermentation of forage products under anaerobic conditions. Moreover, the use of these genera can improve the quality of silage [31]. However, in the current study, the most prevalent genera (Figure 2c) in all the silages were *Klebsiella*, *Enterobacteriaceae*, *Lactobacillus*, and

*Weissella.* This result is slightly different from that of other researchers [32], who found that most bacteria involved in the silage lactic acid fermentation belonged to the *Lactobacillus*, *Pedicoccus*, *Weissella*, and *Leuconostoc* genera. Keshri et al. [29] found that *Pediococcus* grew vigorously at the early silage stage and initiated lactic acid fermentation, thus stimulating the growth of lactic acid bacteria species. Conversely, *Lactobacillus* became more active and grew vigorously with decreasing pH. However, in the present study, the relative abundances of *Klebsiella*, *Enterobacteriaceae*, *Lactobacillus,* and *Weissella* decreased with increasing sucrose addition. Thus, the probable reason for the higher relative abundance of *Klebsiella* and *Enterobacteriaceae* and the lower relative abundance of *Lactobacillus* in this study may be related to the delayed formation of an acidic environment. *Klebsiella*, a member of the *Enterobacteriaceae* family, has been detected in many silage studies [33,34] and competes with lactic acid bacteria for substrates, resulting in a highly acidic environment [35]. The results of this study showed that *Lactobacillus* grows slowly or stagnates because of the low WSC content in kudzu, resulting in the production of harmful bacteria, which may be another reason for the propagation of *Klebsiella.*

Da Silva et al. [36] reported that *Lactobacillus* played an important role in pH reduction during late lactic acid fermentation. In this study, *Lactobacillus* was not the main genus in the kudzu silage, but after 60 d of fermentation, its relative abundance in sucrose-treated kudzu silage was higher than that in the control group. This result showed that sucrose addition could improve the fermentation quality and microbial community of the kudzu silage.

Figure 2 shows that there was a significant interaction between the microbial diversity and fermentation quality of the silage. There was a significant negative correlation between *Streptophyta* and the ADF and CF content, indicating that it contributed to the ADF and CF content in the silage. *Enterococcus* was negatively correlated with the $NH_3$-N content, indicating that it was beneficial for protein preservation.

## 5. Conclusions

Supplementation of kudzu silages with different sucrose concentrations significantly decreased their pH levels, ADF content, and the relative abundance of *Pantoea*. The ensiling duration significantly decreased the content of ADF, CF, and $NH_3$-N and the relative abundance of *Enterobacteriaceae* and *Enterococcus*, while increasing the DM, CP, and DDM contents, and the relative abundance of *Cyanobacteria*, *Actinobacteria*, *Lactobacillus*, *Pantoea*, *Streptophyta*, *Lactococcus,* and *Brevundimonas* in the silages. The RFV values of kudzu silages tended to increase with increasing sucrose content. With the extension of silage time, sucrose addition inhibited harmful bacteria in the kudzu silage and increased the relative abundance of *Lactobacillus* and *Lactococcus*. In conclusion, the results of this study show that sucrose addition can improve the feed quality of kudzu silage.

**Author Contributions:** Conceptualization, Z.H. and D.W.; methodology, D.W.; software, X.Z. (Xuelei Zhang); validation, X.Z. (Xia Zheng), L.Y. and Q.C.; formal analysis, Z.H.; investigation, X.Z. (Xia Zheng); resources, X.Z. (Xia Zheng); data curation, X.Z. (Xuelei Zhang); writing—original draft preparation, Z.H.; writing—review and editing, D.W.; visualization, Z.H. and D.W.; supervision, Z.H. and D.W.; project administration, X.Z. (Xia Zheng); funding acquisition, Z.H. and D.W. All authors have read and agreed to the published version of the manuscript.

**Funding:** D. W. was funded by the Chinese Academy of Agricultural Sciences and Central Public-interest Scientific Institution Basal Research Fund (1610242021005). Z.H. was founded Natural Science Foundation of Hunan Province by Hunan Provincial Department of science and technology (2021JJ50028), and Agricultural Science and Technology Innovation Project Special Fund of Chinese Academy of Agricultural Sciences (ASTIP-IBFC).

**Institutional Review Board Statement:** Not applicable.

**Informed Consent Statement:** Not applicable.

**Data Availability Statement:** The 16S rDNA sequencing data presented in this study are available at the NCBI Sequence Read Archive (SRA) (http://www.ncbi.nlm.nih.gov/sra) (accessed on 6 July 2022) under accession number PRJNA855559.

**Acknowledgments:** The authors would like to thank Xia Yu for the silage sample collection.

**Conflicts of Interest:** The authors declare no conflict of interest. The sponsors had no role in the design, execution, interpretation, or writing of the study.

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
