# Peer review of "Dynamic Profiles of Fermentation Quality and Microbial Community of Kudzu (Pueraria lobata) Ensiled with Sucrose"

_agronomy, doi:10.3390/agronomy12081853_

Round 1

Reviewer 1 Report

Dear, authors:

With some changes, this paper can be improved.

Comments and questions for authors:

In Introduction, for example, there is a reference for Muck et al. in a different format to the rest which are in numbers. Authors must revise it for this and other references in the text.

For me, it is not clear the calculation for RFV.

First paragraph in Results gives the same information that in Table 1, maybe, both are not necessary.

Why is SEM shown only for the first treatment for each item in Table 2? 

When the authors tell about the differences found with Kang et al. in Lines 456-457, they try to explain it saying that in their case these differences existed because the use of kudzu instead of alfalfa with sucrose for silage fermentation. But, exactly which is the reason? Moreover, they had similar results in the growth of the two same phyla.

Can be a problem the production of harmful bacteria (lines 490-491), although the sucrose addition improves the fermentation quality and microbial community in kudzu silage?

Would be the influence of silage time better if it increases more than 60 days?

Respect to others, which are the main novelty of this work?

Author Response

Dear editors and reviewers,

Thank you for your comments concerning our manuscript, “Dynamic profiles of fermentation quality and microbial com-munity of kudzu (Pueraria lobata) ensiled with sucrose” by Zhenping Hou, Xia Zheng, Xuelei Zhang,et al, that we submitted to Agronomy (Article reference: agronomy-1840160).

We found the reviewer’s comments to be very valuable and helpful in improving our presentation, as well as important for guiding significantly to our research. We have read all of the comments carefully and the manuscript have been thoroughly rechecked and provided in the revised manuscript (yellow background) according to these comments and suggestion. Attached please find our revised manuscript, and listed below are our point-by-point response to the reviewer suggestions. Thank you again for handling this manuscript.

With all best wishes,

Yours’ sincerely

Zhenping Hou

Name: Duanqin Wu

E-mail: wuduanqin@caas.cn

Response to Reviewer  Comments

Point 1: In Introduction, for example, there is a reference for Muck et al. in a different format to the rest which are in numbers. Authors must revise it for this and other references in the text.

Response 1: We are very grateful for the reviewer’s hard work. We carefully read the full manuscript, checked all references, unified the annotation way of references in the text, and adjusted their order in the references accordingly.

Point 2: For me, it is not clear the calculation for RFV.

Response 2: We regret that our description failed to make this point clear to reviewer. However, in our materials and methods, all calculation formulas of the relevant data were listed. The calculation formula of RFV is RFV = DMI × DDM× 0.775.

Point 3: First paragraph in Results gives the same information that in Table 1, maybe, both are not necessary.

Response 3: Thanks to the reviewer's suggestions, we carefully reviewed the full text and found that this part had little significance and influence on the full manuscript, so we deleted this part and adjusted the order of tables and other contents in the text accordingly.

Point 4: Why is SEM shown only for the first treatment for each item in Table 2?

Response 4: We thank the reviewer’s hard work. The SEM in all the tables in this manuscript was the total SEM, which was just placed in the first row for the sake of the aesthetics of the table.

Point 5: When the authors tell about the differences found with Kang et al. in Lines 456-457, they try to explain it saying that in their case these differences existed because the use of kudzu instead of alfalfa with sucrose for silage fermentation. But, exactly which is the reason? Moreover, they had similar results in the growth of the two same phyla.

Response 5: We thank the reviewer’s hard work. Indeed, we cite Kang's study to explain these differences existed because the use of kudzu instead of alfalfa with sucrose for silage fermentation. In order to control space, we only scratched the surface of the description, which may confuse the reviewer. In fact, although they had similar results in the growth of the two same phyla, but Kang et al found that the significantly lower Proteobacteria and higher Firmicutes populations in AS2 group than AL group after 60 days offermentation. And what we found that the relative Proteobacteria reached its highest value, while Firmicutes reached its lowest value in the PS2 group after day 60 of ensiling. Our two results were opposite, which was also reflected in the discussion.

Point 6: Can be a problem the production of harmful bacteria (lines 490-491), although the sucrose addition improves the fermentation quality and microbial community in kudzu silage?

Response 6: Thanks to the reviewer. According to 16SrDNA analysis, kudzu silage produced a trace of harmful bacteria during the fermentation process of added sucrose silage, and the specific reasons need to be analyzed in detail. In the process of feeding research on kudzu, we will carry out more in-depth research on the silage processing and utilization of kudzu, so as to minimize or eliminate harmful bacteria before it can be used in animal feeding.

Point 7: Would be the influence of silage time better if it increases more than 60 days?

Response 7: Thanks for the reviewer's suggestion, we will carry out further research on kudzu silage in the future.

Point 8: Respect to others, which are the main novelty of this work?

Response 8: The original intention of this study is to provide a new forage resource and its processing and utilization for herbivore breeders. We hope that this study will provide a reference for the development and utilization of forage resources.

Reviewer 2 Report

In the reviewed paper, the authors were checking if the addition of sucrose to the Kudzu stillage will help to improve the quality of the final product. The submitted manuscript is well prepared and easy to read. The authors present their research methods and obtained results in a clear way, also, in my opinion, the discussion of the presented results is sufficient.

The only thing that should be corrected is in L262 - the authors wrote"The relative Proteobacteria reached its highest value, while Firmicutes reached its highest value in the PS2 group after day 60 of ensiling" - the word "highest" (in bold) should be replaced with "lowest".

Author Response

Dear editors and reviewers,

Thank you for your comments concerning our manuscript, “Dynamic profiles of fermentation quality and microbial com-munity of kudzu (Pueraria lobata) ensiled with sucrose” by Zhenping Hou, Xia Zheng, Xuelei Zhang,et al, that we submitted to Agronomy (Article reference: agronomy-1840160).

We found the reviewer’s comments to be very valuable and helpful in improving our presentation, as well as important for guiding significantly to our research. We have read all of the comments carefully and the manuscript have been thoroughly rechecked and provided in the revised manuscript (yellow background) according to these comments and suggestion. Attached please find our revised manuscript, and listed below are our point-by-point response to the reviewer suggestions. Thank you again for handling this manuscript.

With all best wishes,

Yours’ sincerely

Zhenping Hou

Name: Duanqin Wu

E-mail: wuduanqin@caas.cn

Response to Reviewer  Comments

Point: The only thing that should be corrected is in L262 - the authors wrote"The relative Proteobacteria reached its highest value, while Firmicutes reached its highest value in the PS2 group after day 60 of ensiling" - the word "highest" (in bold) should be replaced with "lowest".

Response: We are very grateful for the reviewer’s affirmation of our work. We carefully checked data and read the manuscript. We found that the input was wrong when writing the original manuscript, which has been corrected in the revised manuscript and marked with yellow background.

Reviewer 3 Report

The abstract and introduction were presented correctly, as well as the methodological part, although some points can be made more precise.

The layout of the work, tables and graphs almost exactly correspond to the work published earlier: Kang et al.: Alfalfa silage treated with sucrose has an improved feed quality and more beneficial bacterial communities. Front. Microbiol. 2021, 12:670165. doi: 10.3389/fmicb.2021.670165. It is, of course, rather a comment about the scope of research conducted by a team of authors from a given research center.

The addition of sucrose was 0.5 and 1% dry or fresh matter?

Line 114: „…and extension (72 °C, 45 s); and extension at 72 °C for 10 min.”Is the last one the final extension?

Line 150-157: a detailed description of the value of the plant material does not seem necessary, as it is an exact repetition of the table contents.

Line 193-194: the sentence is not necessary, it is a repetition of the table title, it is enough to add Table 3 in parentheses in the next sentence after "... pH value". Similarly, the sentence in Line 240-241.

Why reference a table on line 441, „The results in Table 5show that…”can be omitted.

Line 449: there is Kang et al., but [23], which is at the end of this sentence, is missing.

In the discussion, reference was made to only a few issues presented in the cited literature, no more complete explanation of the obtained values was undertaken. The results obtained in comparison with other studies are explained by various methods of ensilage, and yet the basic difference results rather from the value and suitability for ensiling of plant material used in other experiments. Example: the work of Kim et al. (1990) analyzes the effect of starch addition on the value of kudzu silage. This is, of course, work from over thirty years ago and not sucrose but starch was an additive, but maybe it is worth referring to the quality of the plant material?

The work should be assessed positively, but it requires a few indicated minor corrections and additions, which will allow to make a decision about its possible publication.

Author Response

Dear editors and reviewers,

Thank you for your comments concerning our manuscript, “Dynamic profiles of fermentation quality and microbial com-munity of kudzu (Pueraria lobata) ensiled with sucrose” by Zhenping Hou, Xia Zheng, Xuelei Zhang,et al, that we submitted to Agronomy (Article reference: agronomy-1840160).

We found the reviewer’s comments to be very valuable and helpful in improving our presentation, as well as important for guiding significantly to our research. We have read all of the comments carefully and the manuscript have been thoroughly rechecked and provided in the revised manuscript (yellow background) according to these comments and suggestion. Attached please find our revised manuscript, and listed below are our point-by-point response to the reviewer suggestions. Thank you again for handling this manuscript.

With all best wishes,

Yours’ sincerely

Zhenping Hou

Name: Duanqin Wu

E-mail: wuduanqin@caas.cn

Response to Reviewer  Comments

Point 1: The abstract and introduction were presented correctly, as well as the methodological part, although some points can be made more precise.

Response 1: We are very grateful for the reviewer’s affirmation of our work. According to the reviewer's suggestions, we carefully read and deliberated on methodological part. In order to better explain the method adopted in our research process to readers, we only made slight improvements on this part.

Point 2: The layout of the work, tables and graphs almost exactly correspond to the work published earlier: Kang et al.: Alfalfa silage treated with sucrose has an improved feed quality and more beneficial bacterial communities. Front. Microbiol. 2021, 12:670165. doi: 10.3389/fmicb.2021.670165. It is, of course, rather a comment about the scope of research conducted by a team of authors from a given research center.

Response 2: We are from the same team as Kang et al., so we use the same protocol to carry out the research work of the same topic, and have similar habits to display the tables and charts. This magazine has no special requirements for how the data should be presented, so we have compiled these tables and charts according to our team's own habits.

Point 3: The addition of sucrose was 0.5 and 1% dry or fresh matter?

Response 3: The addition of sucrose was 0.5 and 1% fresh weight.

Point 4: Line 114: „…and extension (72 °C, 45 s); and extension at 72 °C for 10 min.”Is the last one the final extension?

Response 4: Thanks for the reviewer's careful reading. We have carefully read the original text. Extension here is indeed “final extension”. This is because the word “final” was accidentally omitted during the writing of the original text.

Point 5: Line 150-157: a detailed description of the value of the plant material does not seem necessary, as it is an exact repetition of the table contents.

Response 5: Thanks to the reviewer's suggestions, we carefully reviewed the full text and found that this part had little significance and influence on the full manuscript, so we deleted this part and adjusted the order of tables and other contents in the text accordingly.

Point 6: Line 193-194: the sentence is not necessary, it is a repetition of the table title, it is enough to add Table 3 in parentheses in the next sentence after "... pH value". Similarly, the sentence in Line 240-241.

Response 6: According to the reviewer's suggestion, we tried to delete redundant sentences to simplify the description of experimental results.

Point 7: Why reference a table on line 441, „The results in Table 5 show that…”can be omitted.

Response 7: Thanks for the reviewer's suggestion. We have re-described this sentence and deleted “Table 5” here.

Point 8: Line 449: there is Kang et al., but [23], which is at the end of this sentence, is missing.

Response 8: Thanks for the reviewer's suggestion. We adjusted the position of the reference notes, and other places in the manuscript were also modified accordingly, marked with blue font and green background.

Point 9: In the discussion, reference was made to only a few issues presented in the cited literature, no more complete explanation of the obtained values was undertaken. The results obtained in comparison with other studies are explained by various methods of ensilage, and yet the basic difference results rather from the value and suitability for ensiling of plant material used in other experiments. Example: the work of Kim et al. (1990) analyzes the effect of starch addition on the value of kudzu silage. This is, of course, work from over thirty years ago and not sucrose but starch was an additive, but maybe it is worth referring to the quality of the plant material?

Response 9: Thanks for the reviewer's comments. This study focused on the effects of sucrose addition on the nutritional quality and microbial diversity of kudzu silage, and silage additive was single, silage method was not very complicated. So we only cited references on the addition of sucrose in different materials in our discussion. In order to control the length of the article, we did not analyze and discuss the experimental results. We will study different silage processing methods of kudzu, and then carry out detailed analysis and discussion.
